# Development of Antimicrobial Blends of Bacteria Nanocellulose Derived from Plastic Waste and Polyhydroxybutyrate Enhanced with Essential Oils

**DOI:** 10.3390/polym16243490

**Published:** 2024-12-14

**Authors:** Everton Henrique Da Silva Pereira, Marija Nicevic, Eduardo Lanzagorta Garcia, Vicente Fróes Moritz, Zeliha Ece Ozcelik, Buket Alkan Tas, Margaret Brennan Fournet

**Affiliations:** PRISM, Research Institute, Technological University of the Shannon, Midlands Midwest, Athlone, Co., Dublin Rd, N37 HD68 Westmeath, Ireland; marija.mojicevic@tus.ie (M.N.); eduardo.lanzagortag@tus.ie (E.L.G.); a00288528@student.tus.ie (V.F.M.); a00314647@student.tus.ie (Z.E.O.); buket.tas@tus.ie (B.A.T.); margaret.brennanfournet@tus.ie (M.B.F.)

**Keywords:** bacterial nanocellulose, polyhydroxyalkanoates, biopolymers, antimicrobial blends

## Abstract

The escalating global concern regarding plastic waste accumulation and its detrimental environmental impact has driven the exploration of sustainable alternatives to conventional petroleum-based plastics. This study investigates the development of antimicrobial blends of bacterial nanocellulose (BNC) derived from plastic waste and polyhydroxyalkanoates (PHB), further enhanced with essential oils. The antimicrobial activity of the resulting BNC/PHB blends was tested in vitro against *Escherichia coli*, *Staphylococcus aureus*, and *Candida albicans*. The incorporation of essential oils, particularly cinnamon oil, significantly enhanced the antimicrobial properties of the BNC/PHB blends. The BNC with 5% PHB blend exhibited the highest antifungal inhibition against *C. albicans* at 90.25%. Additionally, blends with 2% and 10% PHB also showed antifungal activity, inhibiting 68% of *C. albicans* growth. These findings highlight the potential of incorporating essential oils into BNC/PHB blends to create effective antimicrobial materials. The study concludes that enhancing the antimicrobial properties of BNC/PHB significantly broadens its potential applications across various sectors, including wound dressings, nanofiltration masks, controlled-release fertilizers, and active packaging.

## 1. Introduction

The escalating global concern regarding plastic waste accumulation and its detrimental environmental impact has spurred the exploration of sustainable alternatives to conventional petroleum-based plastics [1]. Numerous studies forecast that oil supplies, essential for producing plastic monomers, will be exhausted within this or the next century. As resources become scarce, the urgency for alternative feedstocks for plastic production will become critical, especially given the anticipated continuous growth of the plastics market [2]. Commodity plastics derived from biological resources and bioplastics offer promising alternatives to meet future demands, offering biodegradability and reduced reliance on fossil fuels [3].

Among these, bacterial nanocellulose (BNC) has garnered significant attention due to its remarkable properties, including high purity, mechanical strength, and biocompatibility [4]. Unlike plant cellulose, BNC is highly crystalline and lacks contaminants like lignin and pectin, simplifying purification and enhancing biocompatibility. BNC forms a 3D network of ribbon-like fibers, providing flexibility and moldability for diverse applications, especially in biomedical and food industries where high purity is critical [5,6]. Produced by bacteria such as *Komagataeibacter* and *Acetobacter* species [6], BNCs polymerization and yield depend on factors like strain type, nutrient sources, and production conditions [7]. The commonly used Hestrin-Schramm (HS) medium, however, raises production costs due to its high glucose content. To address these limitations, researchers are actively investigating strategies to optimize BNC production processes, explore cost-effective feedstocks, and modify BNC to improve its properties [8]. One such approach involves incorporating antimicrobial agents into BNC matrices to impart antibacterial and antifungal properties, which are crucial for applications like food packaging and wound dressings [9]. Recent advances also include genetically modified strains to enhance BNC yields and properties, though progress is limited by the availability of genetic tools [10]. Expanding these efforts could enable faster, higher-yield production with additional functionalities to meet industrial demands.

Polyhydroxybutyrate (PHB), a microbial polyester within the PHA family, is one of the most promising biomaterials under study [11]. Produced by various bacterial species, including *Ralstonia eutropha*, *Bacillus* spp., and *Pseudomonas* spp., PHB serves as a carbon and energy reserve under nutrient-limited conditions [12]. While over 75 bacterial genera can accumulate PHB, only a few strains are currently suitable for industrial production due to factors like production cost, yields, and extraction challenges [13]. PHB is known for its high crystallinity, brittleness, and low thermal stability. Its properties, including molecular weight and mechanical strength, are highly influenced by the bacterial strain, carbon source, and culture conditions used [14]. Extraction methods, typically solvent-based, also impact its quality but add to production costs, especially for high-purity medical-grade PHB [15,16]. To enhance PHBs mechanical properties and reduce costs, recent research focuses on optimizing fermentation techniques, modifying material chemistry, and developing copolymers and blends.

Incorporating antimicrobial agents into biopolymers and biodegradable polymers enhances their antibacterial properties, making them ideal for applications like food packaging and medical devices, where preventing microbial growth is essential for safety. These antimicrobial formulations often include metal oxide nanoparticles, such as zinc oxide and copper oxide, natural extracts like turmeric, antibacterial peptides, and other effective compounds [9,17,18]. Essential oils, derived from natural sources, have long been recognized for their antimicrobial properties [19]. These oils contain diverse bioactive compounds that can inhibit the growth of bacteria, fungi, and other microorganisms [20]. Incorporating essential oils into BNC matrices not only enhances their antimicrobial activity but also adds value by utilizing natural and renewable resources.

Building upon our previous work on PET upcycling [21], this study explores the development of antimicrobial blends composed of bacterial nanocellulose (BNC) derived from plastic waste and polyhydroxybutyrate (PHB), enriched with essential oils. While previous research has explored incorporating essential oils into biopolymers for antimicrobial applications [22,23,24,25,26], this study presents a novel approach by utilizing cinnamon oil in BNC/PHB blends derived from plastic waste. This not only imparts antimicrobial activity but also adds value by implementing antimicrobial properties to our plastic waste upcycling strategies [21], addressing the growing need for sustainable materials and effective solutions in combating device-related bacterial infections and mitigating potential health risks [27].

## 2. Materials and Methods

### 2.1. Microorganisms and Culture Condition

*Komagataeibacter medellinensis* ID13488, stored at −80 °C in a modified Hestrin-Schramm (HS) medium containing 25% glycerol, was used for BNC production. Pre-cultures were prepared by transferring aliquots from the frozen stock and incubating them in liquid Hestrin-Schramm medium (HS) (2% glucose, 0.5% peptone, 0.5% yeast extract, 0.27% disodium phosphate, 0.15% citric acid) at 30 °C and 180 rpm for 48 h. For BNC production, borosilicate horizontal trays each containing 1 L of modified-HS medium: post-consumed terephthalic acid (pcTPA) obtained after reactive extrusion (REX) of polyethylene terephthalate was added as a replacement for certain amounts of glucose (1%, *w*/*w*) [21]. Modified HS was further inoculated with 10% *v*/*v* of pre-culture and incubated statically at 30 °C for 14 days.

### 2.2. Antimicrobial Blends Preparation

#### 2.2.1. Essential Oils Selection

All essential oils used in this study were sourced from Atlantic Aromatics, Inc.(Bray, Co. Wicklow, Ireland). To ensure the validity and reproducibility of our study, we conducted a preliminary assessment of their antimicrobial activity. Five essential oils (Cinnamon leaf, Eucalyptus oil, Thyme Linalool, Lemongrass, and Clove bud oil) were tested to evaluate their antimicrobial activity against *E. coli* and *S. aureus*. Cultures were grown overnight in Luria-Bertani (LB) medium (10 g·L-1 casein peptone, 5 g·L-1 yeast extract, 10 g·L-1 NaCl, 15 g·L-1 agar) at 37 °C and 180 rpm. Then, the cultures were diluted to 0.1 OD A630nm, and 100 µL of the diluted cultures were added and spread on LA plates. When the plates dried, three discs were added to each plate, and 5 µL of each essential oil was added on a disc. The plates were incubated overnight at 37 °C. From evaluating and selecting the oil to be used, impregnation in the blends continued.

#### 2.2.2. BNC-Based Blends Preparation

PHB/BNC blends in the form of films were prepared by dissolving 1 g of BNC powder in a solution of PHB and glacial acetic acid through heat and constant agitation for 6 h to achieve a 2, 5, and 10% PHB final composition, called BPHB2, BPHB5, and BPHB10, respectively (Table 1) [28]. The solution was then homogenized using a high-shear mixer (HG-15D, Witeg, Wertheim, Germany) at 17,000 rpm for 60 min with the addition of 1% (*v*/*v*) of selected essential oil. The mixture was then spread onto glass 100 mm × 15 mm petri dishes and subjected to oven drying at 30 °C overnight, resulting in flat, opaque film.

### 2.3. Material Characterization

#### 2.3.1. Blends Morphology

The visual and physical characteristics of the BNC blends were assessed through observation, tactile examination, and 2.5D surface characterization. The samples were visually inspected for color uniformity, surface texture (smoothness or roughness), and any visible defects or irregularities. The tactile assessment involved gently handling the samples to evaluate their flexibility, rigidity, and overall feel. These qualitative observations were recorded and used to correlate with the variations in thermal and chemical properties observed in the further thermogravimetric analysis (TGA) and Fourier-transform infrared spectroscopy (FTIR) analyses.

To complement the qualitative observations made through visual and tactile examination, sample topography was undertaken for those blends exhibiting film-like properties. This involved using 2.5D representation to investigate surface characteristics. Samples were digitized at low magnification using a Stemi 305 stereoscopic microscope equipped with an AxioCam ERc5s camera, and the resulting images were processed using Zen 3.10 Lite software (all from Carl Zeiss GmbH, Oberkochen, Germany).

#### 2.3.2. Scanning Electron Microscopy (SEM)

The surface of the samples was analyzed using scanning electron microscopy (SEM). Back-scattered electron mode images were captured using the Mira XMU SEM (Tescan™, Brno, Czech Republic) with an accelerating voltage of 9 kV. To prepare the samples for SEM imaging, they were placed on an aluminum stub and coated with a thin layer of gold using a Baltec SCD 005 sputter coater (Bal-Tec, Coesfeld, Germany). The sputtering process lasted 110 s and was performed under a vacuum pressure of 0.1 mbar. This coating process was carried out prior to the imaging analysis to ensure optimal sample visualization.

The SEM image processing was performed using PoreSpy v2.4.1, an open-source image library written in Python, using the Sub-Network of an Over-segmented Watershed (SNOW), porosimetry, and local_thickness functions to provide information on the surface morphology and porosity of the BNC blends, including surface heterogenicity and pore size distribution [29].

#### 2.3.3. Tensile Testing

Stress-strain tensile tests were performed with rectangular specimens (25 mm × 8 mm) in a universal testing machine (Zwick GmbH and Co. KG, Ulm, Germany) with a 10 kN load cell and testXpert III software v1.9 by the same manufacturer. The test speed was 2 mm·min^−1^, gauge length was 13 mm, and a 2 N pre-load was applied. Young’s modulus was determined at 0.2% elongation.

#### 2.3.4. Thermogravimetric Analysis (TGA)

Thermogravimetric analysis of dried BNC samples was performed on a Pyris 1 TGA (PerkinElmer, Waltham, MA, USA) to evaluate their thermal properties. Each film (10 mg) was subjected to a temperature range of 25 to 800 °C at a heating rate of 10 °C·min^−1^ in order to obtain thermogravimetric curves.

#### 2.3.5. Fourier Transform Infrared Spectroscopy Analysis (FT-IR)

FT-IR analyses were performed using a Perkin-Elmer Spectrum One FT-IR spectrometer (Perkin Elmer Inc., Washington, DC, USA) fitted with a universal ATR sampling accessory and Perkin Elmer software v 10.4.3, which was used to record the spectra of dried BNC films, applying 4 cm^−1^ and 20 scans resolution on a 4000–650 cm^−1^ spectrum.

### 2.4. Antimicrobial Assessment

In this study, the focus was on comparing the antimicrobial effects of different essential oils and BNC/PHB blends. Therefore, the disc diffusion assay was used as a complementary method to the MIC experiment, and the microdilution assays were conducted without a control sample (no essential oil or blend). The results are presented as a comparison between the different essential oils or blends.

#### 2.4.1. Minimum Inhibitory Concentration (MIC)

The antimicrobial activity of six essential oils—cinnamon leaf (Cin); eucalyptus (Euc); thyme linalool (Thy); lemongrass (Lem); clove bud oil (Clo); and ginger oil (Gin)—was evaluated against three microbial strains: *E. coli*, *S. aureus*, and *C. albicans*. Cultures of the strains were incubated overnight at 37 °C and 150 rpm in LB for the bacteria and Sabouraud medium (SAB: 40 g·L^−1^ dextrose, 10 g·L^−1^ peptone, 20 g·L^−1^ agar, pH 5.6.) for *C. albicans*. The minimum inhibitory concentration (MIC) was determined using a broth microdilution method in 96-well plates. Essential oils were initially diluted 2.5-fold in sterile DMSO (400 µL oil + 600 µL DMSO). Each well contained 100 µL of sterile LB media with serial two-fold dilutions of the oils. Microbial suspensions were standardized to an optical density (OD) of 0.01 at 600 nm for bacterial strains and 530 nm for *C. albicans*. Each well received 100 µL of the standardized microbial suspension, except for negative controls. Plates were incubated under static conditions at 37 °C for 24 h. Growth inhibition was assessed by measuring absorbance using a plate reader. Experiments were performed in triplicate. Complementary analysis was performed using the disc diffusion method. Overnight bacterial cultures were diluted to OD_600_ = 0.01 and spread on LA plates. Sterile discs were placed on the inoculated plates, and 5 µL of each essential oil was applied per disc. Plates were incubated at 37 °C for 24 h.

The antimicrobial activity of the resulting BNC/PHB blends was tested using the same procedure: Circular pieces of size 5 mm in diameter of the blended films were cut and sterilized under a UV lamp for 30 min each side. The sterilized films were immersed in the inoculated wells (except the wells assigned for microbial growth control (+)) and incubated overnight at 37 °C under static conditions. Afterward, the films were carefully removed, and the OD was read in a spectrophotometer plate reader at A_630nm_ for the bacteria and A_530nm_ for *C. albicans*. The growth percentage of each well was calculated based on the average OD from the microbial growth wells, which are considered to be 100% growth.

#### 2.4.2. Determination of Biofilm Inhibition Activity

The biofilm inhibition activity of the BNC blends was determined using the crystal violet (CV) assay method [30] against *E. coli*, *S. aureus*, and *C. albicans*. Each cell culture (10% microbial inoculum/medium, 0.5 MacFarland) was added to wells containing the growth medium and BNC/PHB blends (as sheets) and incubated at 37 °C for 48 h to determine the biofilm inhibition activity. Following incubation, the samples were removed from the wells, and the wells were washed with PBS (0.01 M, pH 7.4) to remove non-adherent cells. Each well was then stained with 0.1% CV for 15 min. Subsequently, 33% glacial acetic acid was added to the wells containing *S. aureus* and *C. albicans*, and 95% ethanol was added to the wells containing *E. coli*. The optical density of the wells was measured at 570 nm using a microplate spectrophotometer (Epoch, Biotech, East Falmouth, MA, USA). All tests were performed in triplicate. The following formula was used to calculate the biofilm inhibition percentage of the blends:Biofilm inhibition (%) = [(Control OD − Sample OD)/Control OD] × 100.

To ensure a clear understanding of the antimicrobial activity, only positive results were considered valid and used for further analysis. This means that only instances where the calculated biofilm inhibition percentage was greater than zero were included in the evaluation of the BNC/PHB blends’ efficacy.

#### 2.4.3. Caenorhabditis Elegans Survival Assay

The nematode *Caenorhabditis elegans* AU37 (glp-4; sek-1), obtained from the Caenorhabditis Genetics Center (CGC), University of Minnesota, Minneapolis, MN, USA, was used to establish the toxicity of BNC/PHB materials.

M9 medium was used as the main solvent for sampling, which consisted of 45 mL of 5 × M9 salt, 50 mL of glucose solution 50% (*w*/*v*), 500 µL of 1 M MgSO_4_, 25 µL of 1 M CaCl_2_, 250 µL of vitamins, 250 µL of trace elements solution, 125 µL of ampicillin 50 µg·mL^−1^, and a final volume adjusted to 250 mL with double distilled water.

For this assay, 2 mg of samples were suspended in M9 in triplicates and incubated at 37 °C with orbital agitation at 180 rpm for 48 h. Further, specimens were removed from the suspension, remaining the extracts labeled after the objects: BNC, BPHB2, BPHB5, BPHB10, PHB, EoC (cinnamon oil), two 1:2 serial dilutions, EoD1 and EoD2, respectively.

The worm was propagated under standard conditions, synchronized by hypochlorite bleaching, and cultured on a nematode growth medium using *E. coli* OP50 as a nutrient source [31]. The *C. elegans* AU37 survival assay followed the standard procedure [32] with some minor modifications. Briefly, synchronized worms (L4 stage) were suspended in a medium containing 95% M9 buffer (3.0 g of KH2PO4, 6.0 g of Na2HPO4, 5.0 g of NaCl, and 1 mL of 1 M MgSO4∙7H2O in 1 L of water), 5% LB broth (Oxoid, Basingstoke, UK), and 10 μg·mL^−1^ of cholesterol (Sigma-Aldrich, Munich, Germany). The experiment was carried out in 96-well flat-bottomed microtiter plates (Sarstedt, Nümbrecht, Germany) with a final volume of 100 µL per well. The wells were filled with 50 µL of the suspension of nematodes (25–35 nematodes) and 50 µL of tested BC film suspension. Subsequently, the plates were incubated at 25 °C for 72 h. The fraction of dead worms was determined by counting the number of dead worms and the total number of worms in each well using a stereomicroscope (SMZ143-N2GG, Motic, Wetzlar, Germany). As a negative control experiment, nematodes were exposed to the medium containing 1% DMSO.

Data analysis was performed using the lifelines library in Python 3 [33] to perform Kaplan–Meier survival analysis. Furthermore, the script performed a log-rank test to statistically compare the survival curves of each group to the control, providing *p*-values to indicate significant differences in survival.

## 3. Results and Discussion

### 3.1. Essential Oils Selection

#### Essential Oils Minimum Inhibitory Concentration (MIC)

The initial assessment of the antimicrobial potential of the essential oils against *E. coli*, *S. aureus*, and *C. albicans* was conducted using the broth microdilution method to determine the MIC and the disc diffusion to assess the zones of inhibition as a complementary method. The MIC values, as shown in Figure 1, revealed distinct antimicrobial activities among the tested essential oils. Cinnamon leaf (Cin) exhibited the lowest MIC (22.5 mg·mL^−1^) against *E. coli* and *S. aureus*, indicating its potent antimicrobial efficacy, as in the available literature [34]. Eucalyptus (Euc), Thyme linalool (Thy), and Lemongrass (Lem) displayed moderate MICs ranging from 90 to 180 mg·mL^−1^, suggesting their moderate inhibitory effects. Clove bud oil (Clo) and Ginger oil (Gin) showed the highest MICs (>180 mg·mL^−1^), indicating their relatively weaker antimicrobial activity.

The disc diffusion assay, as shown in Figure 2, further confirmed the antimicrobial potential of the essential oils. Cin and Clo consistently produced large zones of inhibition against both *E. coli* and *S. aureus*, reinforcing their strong antimicrobial effects. Euc and Thy exhibited moderate zones of inhibition, while Lem and Gin showed smaller zones, indicating their weaker activity.

The quantitative analysis of the zones of inhibition, presented in Figure 3, provides a clearer comparison of the antimicrobial effects. Cin exhibited the largest inhibition zones against both *E. coli* and *S. aureus*, with a median zone of 10.6 mm against *S. aureus*. The box plot distribution also highlights the consistency of Cin’s antimicrobial activity.

The selection of essential oils for further investigation was based on their MICs and zones of inhibition. Cin, with its potent antimicrobial activity and consistency, was chosen as the primary essential oil for incorporation into the BNC blends. These results were also reported in the literature. For instance, a study by Burt (2004) [19] demonstrated the effectiveness of cinnamon oil in inhibiting the growth of various microorganisms, including *E. coli* and *S. aureus*.

### 3.2. Blends Morphology

Figure 4 provides a visual representation of the six prepared samples, each with varying compositions of BNC, PHB, and PHA-enriched biomass. The BNC/PHB blends (BPHB2, BPHB5, BPHB10) exhibited distinct visual characteristics depending on the PHB concentration. BPHB2, with the lowest PHB content (2%), displayed a smooth, glossy surface and a homogeneous appearance. As the PHB concentration increased in BPHB5 (5%) and BPHB10 (10%), the surfaces became rougher and more irregular, indicating potential changes in the mechanical properties.

The selection of 2%, 5%, and 10% PHB for the BNC/PHB blends represents an initial exploration into the effects of PHB incorporation on BNC-based materials, with BNC serving as the primary composite component rather than an additive. This approach aims to enhance the value of upcycled plastic waste and result in a relevant biomaterial with improved properties. Further optimization studies are needed to fully understand the influence of PHB concentration on the final material properties and to determine the optimal composition for specific applications.

The BNC/PHA blends (BPST, BPSE, BPSW) also showed variations in their morphology. BPST exhibited a uniform light brown color and a smooth texture, while BPSE had a granular, rough texture. BPSW presented a flat and polished surface with light-reflecting properties. The observed differences in texture and color suggest variations in the distribution and interaction of the components within the blends. However, all the BNC/PHA blends were found to be brittle and fragile, making them unsuitable for further investigation and removed from this work.

To further investigate the surface characteristics of the BNC/PHB blends, topographic analysis was conducted, as shown in Figure 5. The 2.5D topographic simulations revealed that the modifications significantly altered the material’s color, changing from the natural yellow of BNC to brown shades. This color change could be attributed to the increased thickness of the modified samples, which affects their optical properties (including absorption, reflection, and transmission).

The topographic analysis also highlighted differences in surface roughness between the treated samples and the control sample (pure BNC). The modifications resulted in coarser surfaces with irregular topography and variable roughness, in contrast to the smooth surface of the control sample. These observations are consistent with the visual and tactile assessments.

These visual and tactile differences correlate with the variations in thermal and chemical properties observed in the TGA and FT-IR analyses, demonstrating how additive concentrations and processing conditions impact the physical properties of the BNC blends.

As observed by Fernandes et al. [35], the integration of PHB with cellulose networks creates a denser fiber structure with enhanced mechanical strength, resulting from the strong interfacial adhesion and interactions between cellulose chains and PHB; however, another study reported that tensile strength decreases when PHB content exceeds 50% in the BNC membrane, due to its inherent stiffness and brittleness, which contribute to its poor mechanical properties [36]. To date, there has been no significant progress or reported findings in the literature regarding biomass blending with BNC, to the best of the author’s knowledge. The only comparable study performed by Mautner et al. [37] examined substrates for printed electronics and filter nano-papers for water treatment by mixing traditional BNC with fibers derived from softwood pulp cellulosic biomass, which allowed for more effective utilization of the exceptional properties of these fibrils.

### 3.3. Scanning Electron Microscopy (SEM)

The SEM images (A to D) in Figure 6 illustrate the surface morphology of the BNC blends with varying compositions. The subplot Figure 6A (pure BNC) shows a surface with a few wrinkles and highly porous, clear characteristics of pure BNC fibers 3D-networking in conjunction with bacterial cellular structure. Such characteristics are supported by the SNOW segmentation algorithm contrast color (Figure 6E), reflecting higher peaks and valleys heterogeneity [38], and the average pore cumulative distribution when compared to the other materials (Figure 6I). Subplot Figure 3B (BPHB2) indicates 2% PHB content and BNC result in a rougher surface with more pronounced wrinkles and pores differences and dimensions, in contrast to the 5% PHB composition blend (BPHB5), which indicates the chemical interaction sites are better distributed and established in these contents, reflecting a more homogeneous and less porous surface than BNC, qualitatively and quantitatively confirmed with the SNOW and pore size distribution analyses, respectively. The subplot Figure 6D, related to the BPHB10, demonstrates that an increase in the PHB content generates an excess of available sites, reflecting on a more heterogenic matrix, especially close to the surface and pore size distribution properties of pure BNC.

### 3.4. Tensile Properties

Tensile testing results indicate that the treatment had a weakening effect on the mechanical properties evaluated, particularly on the Young’s modulus at 0.2% strain (E). At a concentration of 2% PHB, E drops 46%, then, at 5% PHB, rising to 1173.7 MPa and reducing again to 657.4 MPa at 10% PHB (Figure 7), and are summarized on Table 2.

The same trend was observed for ultimate tensile strength (UTS) and stress at break (σ_B_), with the sample treated at 5% PHB achieving the highest values amongst the modified samples, reaching 56% of the original figures. BPHB10 displayed the lowest values of UTS at 6.9 MPa, a decrease of 75% and 55% relative to control and BPHB5, respectively. The diminished UTS and σ_B_ in the modified BNC samples indicate a reduction in both resilience and toughness compared to the control sample. This implies a decreased capacity of the material to absorb energy during deformation, leading to earlier failure shortly past UTS, suggesting an impaired performance within the plastic regimen, i.e., as BNC underwent modification by the incorporation of PHB, it developed a brittle characteristic under stress.

However, tensile tests were performed with a sparse volume of specimens due to sampling constraints, which imposed restrictions that limited the extent of this analysis. This caused the *p*-value to exceed the conventional threshold of 0.05 in most cases, indicating that the observed differences in group means are short of sufficient statistical evidence to be deemed significant and are statistically indistinguishable based on the available data. Nonetheless, according to Fisher’s LSD post hoc test, BNC and BPHB10 were found to be significantly different (*p* < 0.05).

The changes observed in mechanical properties are likely associated with the morphological features of the samples (Figure 6). The rougher surface with more pronounced wrinkles and fewer pores in BPHB2 could reduce the mechanical performance relative to BNC. Increasing the PHB content to 5% favored the formation of chemical interaction sites, well dispersed within the matrix, and the recovery of microfibrils, hence possibly promoting E, UTS, and σ_B_ to display some recovery. On the other hand, for BPHB10, further increasing the interaction sites leads to a heterogeneous morphology, presumably reducing the load absorption capacity of the matrix and intensifying the crack growth and propagation. A decrease in E and UTS is a consequence of poor dispersion of PHB within the BNC matrix, resulting in a less dense fiber structure [39]. It has been proposed that BNC is of potential interest for improving the mechanical performance of PHB, which has been deemed insufficient in spite of its attractive characteristics such as renewable sources and biodegradability [39].

### 3.5. Thermogravimetric Analysis (TGA)

To provide a comprehensive analysis of the thermal behavior of the BNC blends, thermogravimetric analysis (TGA) was conducted. The TGA curves, depicted in Figure 8, reveal thermal behavior pronounced by similarity. When comparing the TGA curves of BNC and the composites of BNC with varying concentrations of PHB, the pure BNC sample exhibits an initial minor mass loss from room temperature to around 200 °C due to the evaporation of water molecules. The major degradation event for all samples occurred between approximately 200 °C and 400 °C, corresponding to the partial oxidative decomposition of the BNCs carbohydrate segment. The addition of PHB to BNC reflects a minor decrease in the onset temperature of degradation (no significant impact on thermal stability), while the blend BPHB10 presents a more significant difference from T_d,%50_, where the mass loss demands more energy, as illustrated by the purple curve difference. The residual mass at 800 °C decreases with PHB contents of 2% and 10%.

### 3.6. Fourier Transform Infrared Spectroscopy Analysis (FT-IR)

Due to the compound’s composition differences and for better comparative visualization, two groups were also separated for FT-IR profile comparison. The BNC spectrum (Figure 9), which is used as a reference for both panels, displays characteristic peaks at 3350 cm^−1^ (OH stretching), 2922 cm^−1^ (CH stretching of CH2 and CH3 groups), 1625 cm^−1^ (OH deformation), and 1056 cm^−1^ (anti-symmetric out-of-phase bending [40]. These peak profiles remain consistent in the other composite materials, as BNC is the most abundant compound in all the blends.

Focusing on the first panel Figure 9. As the concentration of PHB increases, new peaks emerge, particularly around 1625 cm^−1^, attributed to vibrations of aromatic ring C = C of amide in PHB, and 1054 cm^−1^ [41]; the same occurs with the main BNC peaks, such as the region around 3350^−1^ (H-bond) and 2922 cm^−1^ (CH stretching of CH2 and CH3 groups). The intensity of the PHB peaks increases with higher PHB content, as expected, which confirms the successful incorporation of PHB into the BNC matrix. However, the transmittance intensity of the peaks around 1056 cm^−1^ (anti-symmetric out-of-phase bending) of BPHB5 (~50%) differs non-proportionally with BPHB2 (~76%) and BPHB10 (~66%).

### 3.7. Antimicrobial Activity

The films were subjected to micro-cultivations of *E. coli*, *S. aureus*, and *C. albicans*. Figure 10 illustrates the antimicrobial effect of the composites co-produced with cinnamon oil against these microorganisms. Results are expressed as a percentage relative to the positive control (PC), which represents the growth of microorganisms without any antimicrobial agent. Against *E. coli*, only the PHB composites, BPHB2 and BPHB10, exhibited significant growth inhibition, around 38% and 32%, respectively. Following bacterial gram-positive (*S. aureus*) and fungal (*C. albicans*) growth inhibition, the blends composed of PHB exhibited the highest inhibition; on *S. aureus*, the effect of all BNC-PHB samples was not statistically different, representing an average inhibition of 49%. Besides, the antifungal effect was the one to be highlighted; BPHB5 exhibited the highest inhibition against *C. albicans* (90.25%) with a minor standard deviation of ±0.95, followed by BPHB2 with no statistical difference but a higher deviation (SD ± 6.85), and BPHB10 with around 68% growth inhibition. As per literature, the BC/PHB blend activated with clove essential oil proved that the oil started migrating within the first 6 h of contact with the bacterial cells, resulting in an approximately 65% reduction in *Escherichia coli* UCP 1517223.

In terms of biodegradation, crucial to this kind of material feasibility, considering that modern biomaterials often include additional compounds such as other biopolymers, additives, and stabilizers, the ability of the strains described in Herrera et al. [42] to degrade BNC/PHB (40%) blends was evaluated using *Streptomyces* sp. DG19 and a soil burial assay by Herrera et al., observing 85% weight loss after two weeks. Previous studies have indicated that increasing the BNC content in the blend enhances degradation in vitro [43]. Similar findings were observed in a study involving biocomposites reinforced with nanocellulose in a polyvinyl alcohol matrix during soil burial assays [44]. The results suggest that the above-mentioned strains could potentially be applied to degrade more complex biomaterials that include various additives and plasticizers. This indicates BNC/PHB and BNC/PHA blends would be biodegradable. This hypothesis will be addressed in further investigations to assess the biodegradability of the blends and their potential applications in various fields.

### 3.8. Determination of Anti-Biofilm Activity

Analysis of anti-biofilm activity showed varying inhibitory effects of the compositions against established biofilms of *E. coli*, *S. aureus*, and *C. albicans* (Figure 11). Control (pure bacterial nanocellulose) and 2- and 5% compositions inhibited *E. coli* biofilm formation to different extents. Thus, control should be established without the presence of an antimicrobial agent (essential oil). The BNC, however, exhibited a high virtual level of anti-biofilm activity against *E. coli* on this experimental method. Such divergent effects on *E. coli* growth were reported with different types of nanocellulose [45], which can be related to the structural properties of BNC since its function as biofilm provides a protective environment for bacteria, shielding them from environmental stress, which promotes an alternative matrix for *E. coli* adherence/growth [45,46]. BPHB2 and BPHB5 also demonstrated biofilm inhibitory effects against *E. coli*; despite BPHB2s unclear divergent effect from BNC content on *E. coli*, it demonstrated clear antimicrobial inhibition against *C. albicans*, supporting previous planktonic antimicrobial results (Figure 10). BPHB5s effect, while exhibiting a more homogeneous and significantly smaller porous radius, reveals a potential difference in efficacy against planktonic cells versus biofilm-associated cells, based on strong activity against planktonic cells and reduced activity against biofilms, particularly with *S. aureus*, as no anti-biofilm activity was detected, indicating that the compositions are not effective in disrupting or preventing the formation of *S. aureus* biofilms. This highlights the challenge of targeting biofilms, as the extracellular matrix and altered physiology of biofilm cells can confer increased resistance.

The antimicrobial activity of the BNC/PHB blends may be influenced by the structure of the microorganisms and the physicochemical properties of the samples. Gram-negative bacteria, such as *E. coli*, have an outer membrane that can act as a barrier to antibiotic agents. Gram-positive bacteria, such as *S. aureus*, lack this outer membrane, which may make them more susceptible to the antimicrobial effects of the BNC/PHB blends. The BNC, however, exhibited high virtual levels of anti-biofilm activity against *E. coli* in this experimental method. Such divergent effects on *E. coli* growth can be related to the structural properties of BNC, since its function as a biofilm provides a protective environment for bacteria, shielding them from environmental stress, which promotes an alternative matrix for *E. coli* adherence/growth.

The specific mechanisms underlying the difference in antimicrobial activity against Gram-negative and Gram-positive bacteria are still unclear. Further research is needed to fully elucidate the complex interactions between the BNC/PHB blends, essential oils, and microbial cell structures [47].

### 3.9. Caenorhabditis Elegans Survival Curves

The toxicity of the BNC/PHB materials was evaluated using the *C. elegans* AU37 survival assay, as shown in Figure 12A. Survival probability at 72 h was impacted by exposure to the different materials. While most materials showed a slight reduction in survival compared to the control, EoC (cinnamon undiluted essential oil) exhibited a marked decrease in survival probability (approximately 50%). In Figure 12B, the significance is highlighted by a log-rank test, revealing a statistically significant difference in survival between the EoC group and the control group (*p* < 0.05). This indicates that exposure to cinnamon oil significantly reduced the survival of *C. elegans* compared to the control. However, neither the oil dilutions nor blends showed a statistically significant difference in survival compared to the control. These findings suggest that the antimicrobial activity detected on the liquid assay does not exhibit analogous toxicity.

## 4. Conclusions

This study successfully developed antimicrobial blends of BNC and PHB derived from plastic waste, enhanced with essential oils. The incorporation of cinnamon oil significantly improved the antimicrobial activity of the BNC/PHB blends against *E. coli*, *S. aureus*, and *C. albicans*. Notably, the BPHB5 blend demonstrated the highest antifungal activity against *C. albicans* (90.25% inhibition), while still reflecting moderate activity against the bacterial cultures. These findings highlight the potential of utilizing BNC/PHB blends as sustainable and effective antimicrobial materials.

Future research will focus on optimizing the antimicrobial properties of these blends through methods like soaking or double layering, exploring their efficacy against a wider range of microbial strains, and as carriers for other antimicrobial compounds. The versatility and biodegradability of these materials make them promising candidates for various applications, including wound dressings, nanofiltration masks, controlled-release fertilizers, and active packaging, contributing to a more sustainable and healthier future.

## Figures and Tables

**Figure 1 polymers-16-03490-f001:**
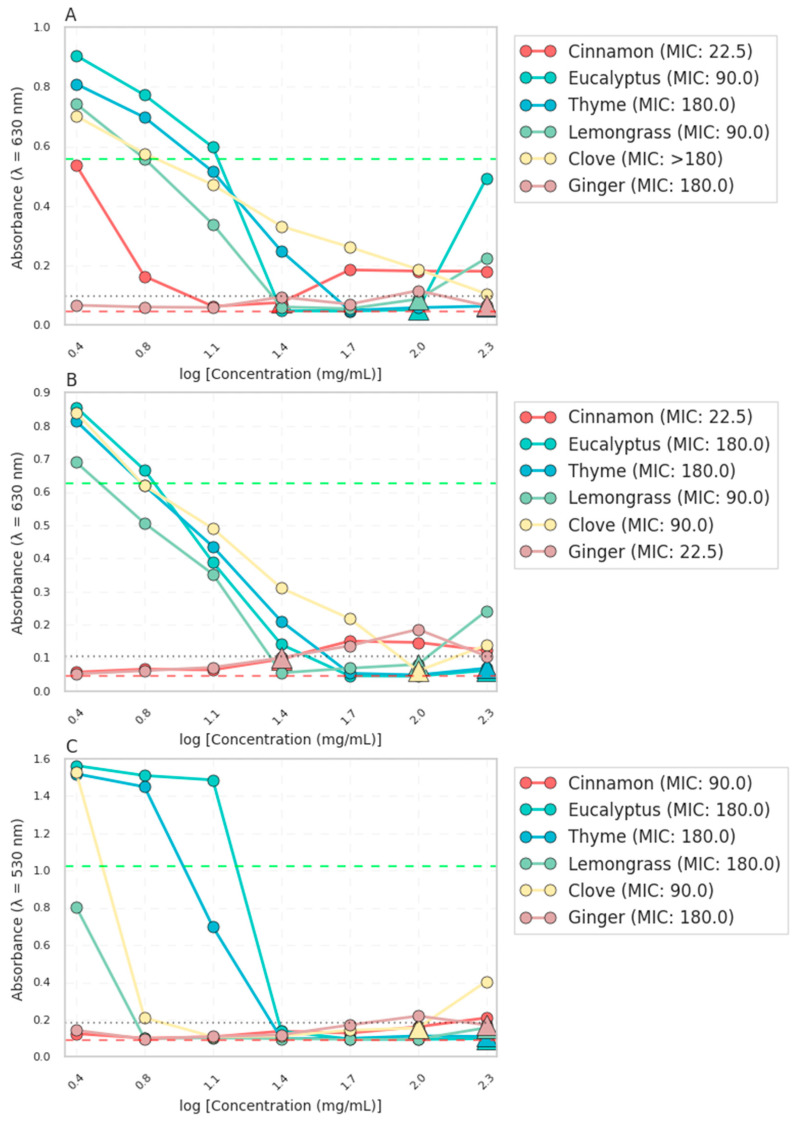
Antimicrobial activity of essential oils against (**A**) *E. coli*, (**B**) *S. aureus,* and (**C**) *C. albicans* in liquid culture. The absorbance values were measured at 530 nm for *S. aureus* and 630 nm for *E. coli* and *P. aeruginosa* after 24 h of incubation. The dashed lines represent the positive (red) and negative (green) controls, while the dotted line represents the MIC threshold, and triangles indicate the minimum inhibitory concentration (MIC) for each essential oil.

**Figure 2 polymers-16-03490-f002:**
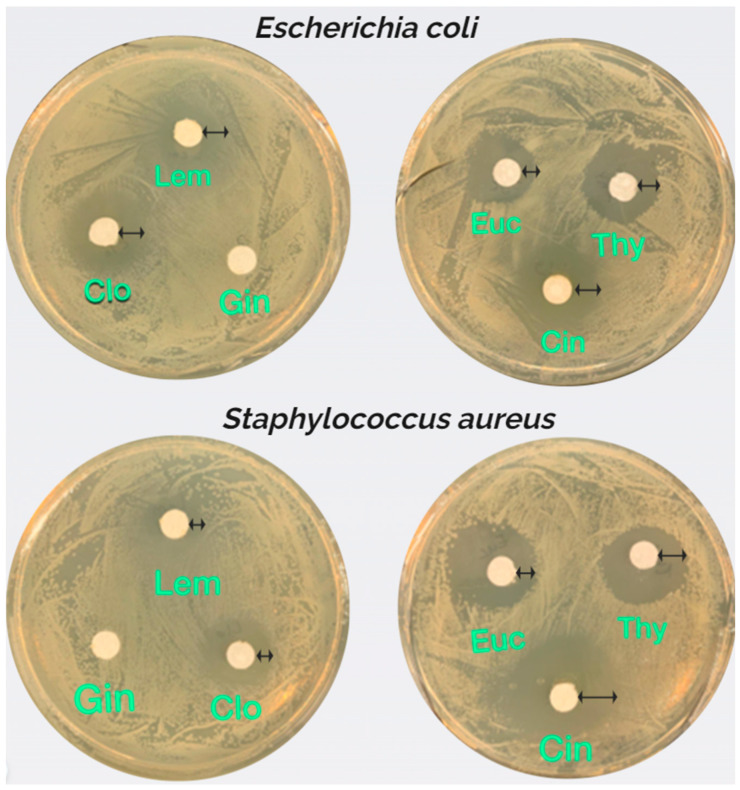
Zones of inhibition of examined essential oils against *E. coli* and *S. aureus* in a Ø 89.42 mm standard petri dish. Arrows represent the halo’s radius.

**Figure 3 polymers-16-03490-f003:**
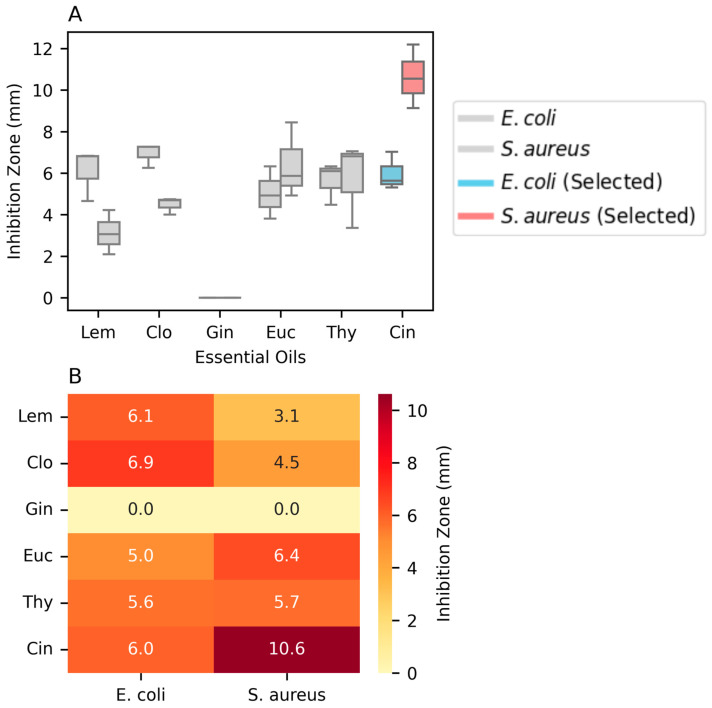
Antimicrobial activity of essential oils against *E. coli* and *S. aureus*. (**A**) Box plot of the antimicrobial activity of the essential oils (**B**) Heatmap of the antimicrobial activity of the essential oils against *E. coli* and *S. aureus*.

**Figure 4 polymers-16-03490-f004:**
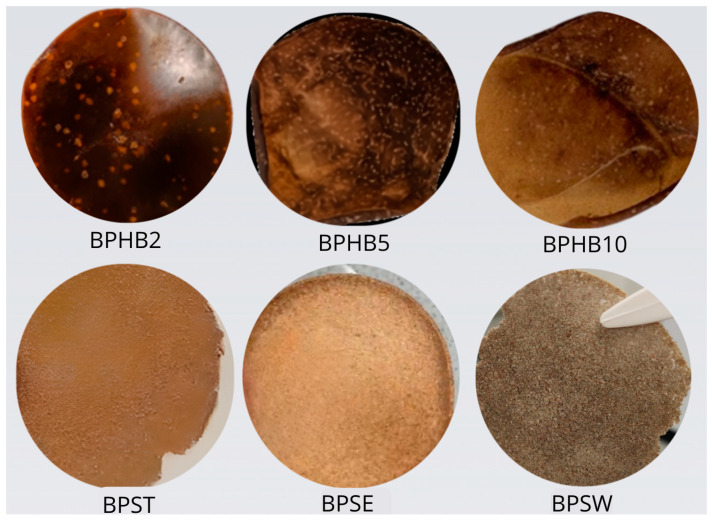
Visual and Physical Characteristics of BNC Blends with PHB (**top**) and PHA-enriched biomass (**bottom**), in a Ø 89.42 mm standard petri dish.

**Figure 5 polymers-16-03490-f005:**
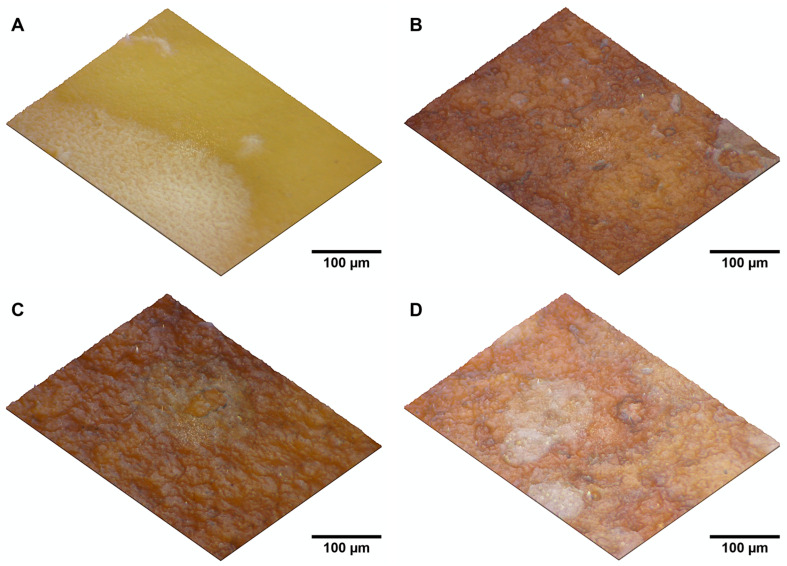
Materials 2.5D topographic simulations at 15× magnification of BNC (**A**), BPHB2 (**B**), BPHB5 (**C**), and BPHB10 (**D**).

**Figure 6 polymers-16-03490-f006:**
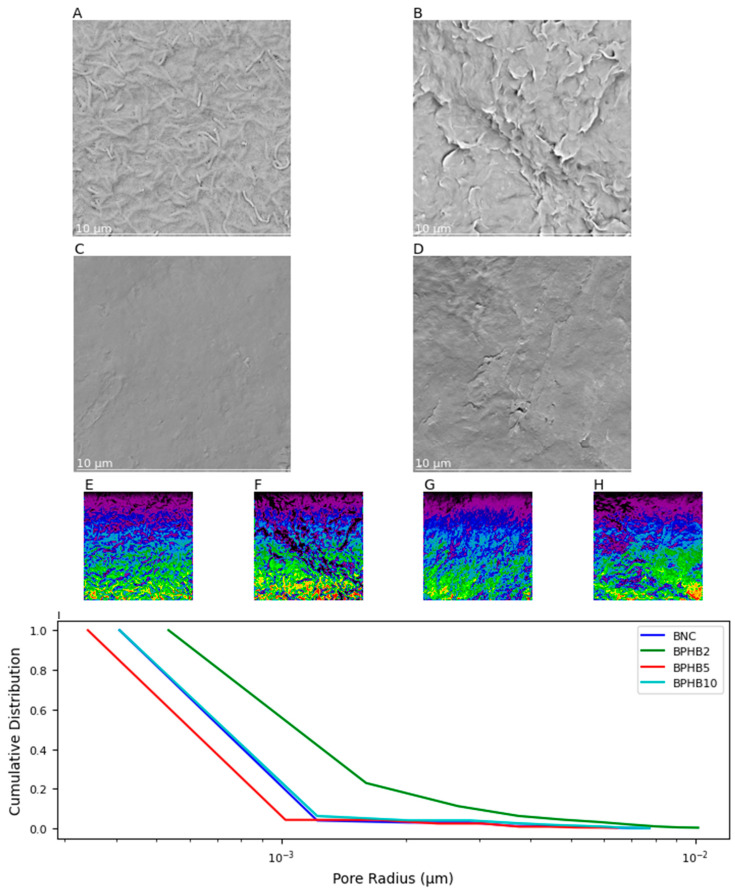
SEM images and pore size distribution analysis of BNC/PHB blends. (**A**) BNC, (**B**) BPHB2, (**C**) BPHB5, and (**D**) BPHB10. SNOW algorithm segmentation of (**E**) BNC, (**F**) BPHB2, (**G**) BPHB5, and (**H**) BPHB10 from same figure from (**A**–**D**), respectively, which the color gradient visualizes pore size variation within and between materials red (largest pores) to blue/purple (smallest); (**I**) Cumulative pore size distributions of BNC/PHB blends.

**Figure 7 polymers-16-03490-f007:**
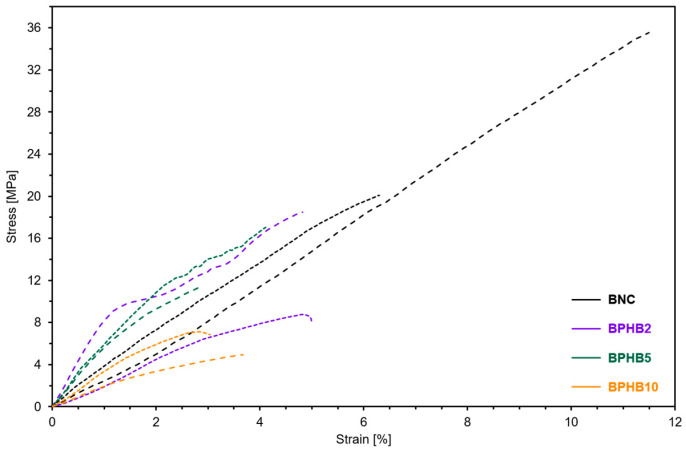
Examples of stress-strain tensile curves obtained, grouped by color and duplicates differentiated by dashed and dotted lines.

**Figure 8 polymers-16-03490-f008:**
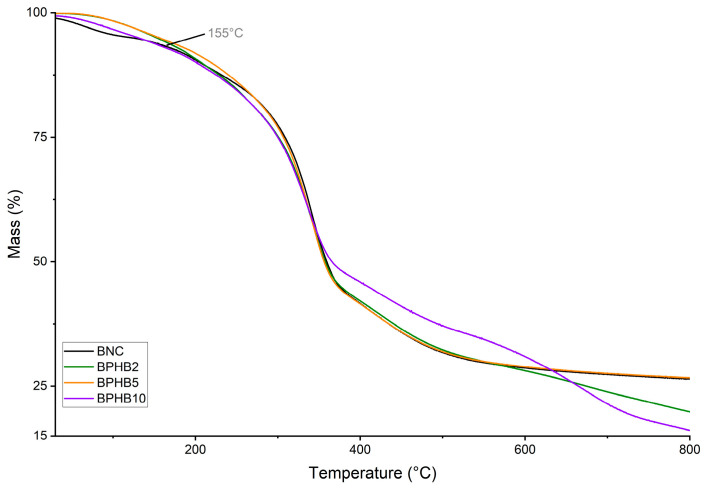
TGA curves of examined BNC blends: BNC/PHB blends.

**Figure 9 polymers-16-03490-f009:**
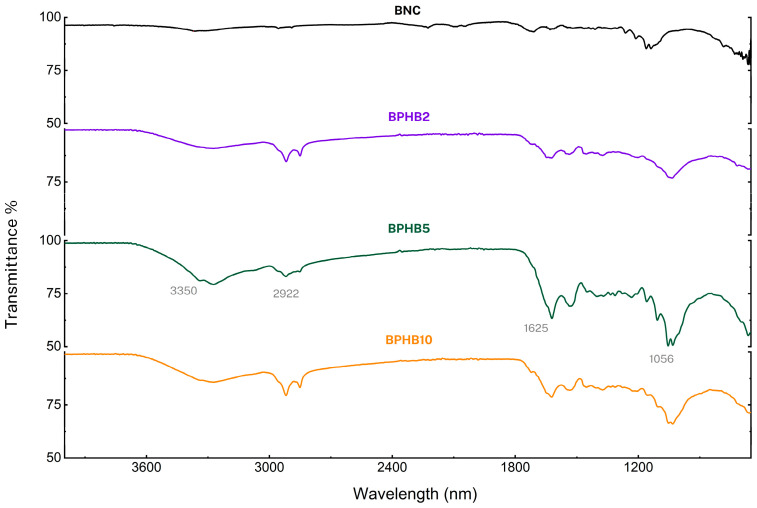
FT-IR spectra of examined BNC blends.

**Figure 10 polymers-16-03490-f010:**
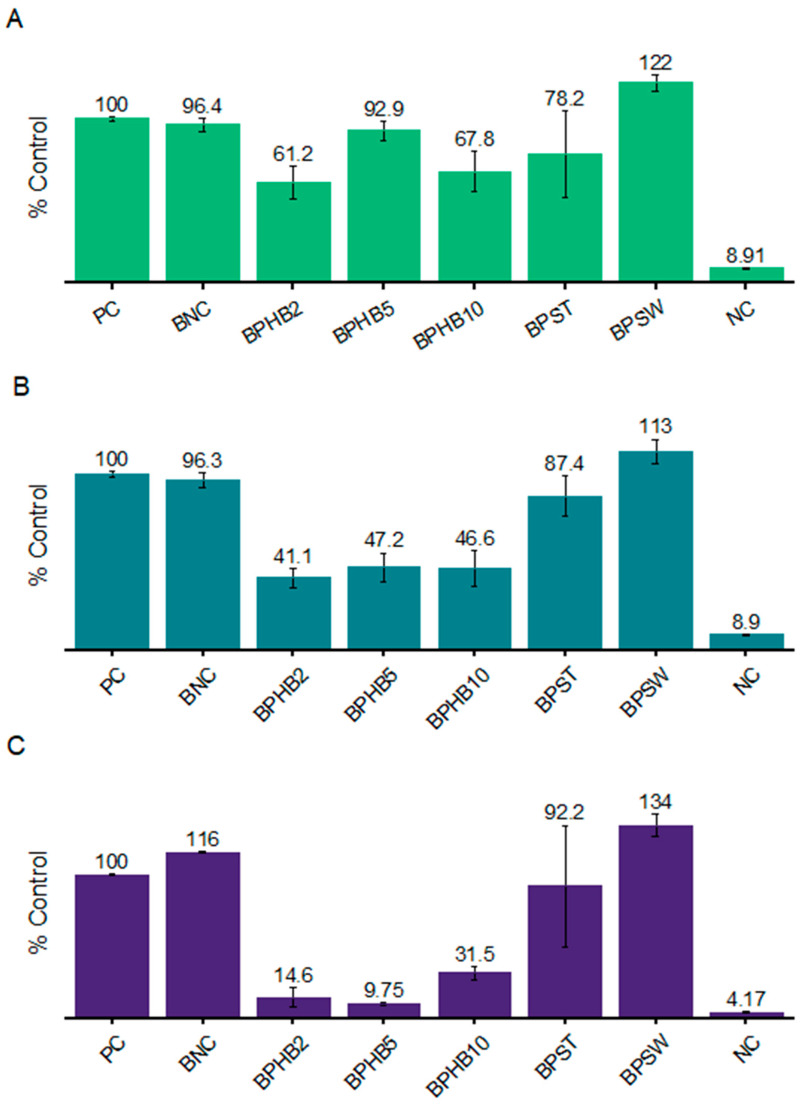
Microbial growth graph depicting cultures of *E. coli* (**A**), *S. aureus* (**B**), and *C. albicans* (**C**) as positive controls, alongside cultures growing in the presence of BNC-based blends with and without the addition of Cinnamon oil. Growth percentages are calculated relative to the average growth of the positive controls, set at 100%.

**Figure 11 polymers-16-03490-f011:**
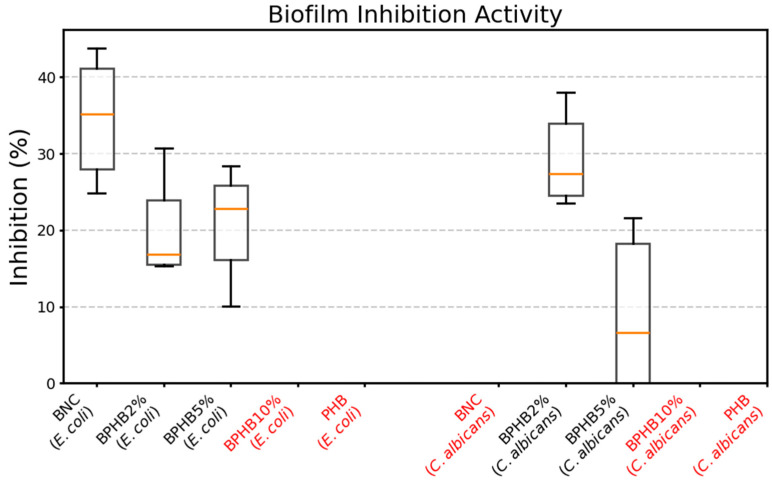
Effects of Bacterial Nanocellulose and BNC-PHB Composites on Microbial Biofilms. The samples in red indicate the absence of anti-biofilm activity.

**Figure 12 polymers-16-03490-f012:**
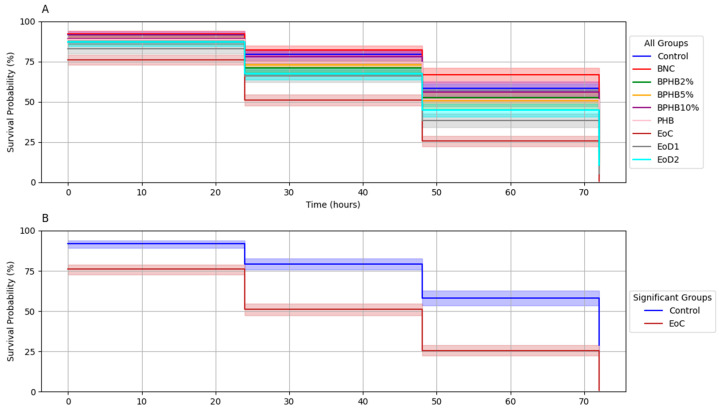
Survival analysis of *C. elegans* AU37 exposed to BNC/PHB materials. Kaplan–Meier survival curves for (**A**) all experimental groups and (**B**) groups with statistically significant differences.

**Table 1 polymers-16-03490-t001:** Compositions of prepared blends.

Composition	Abbreviations
2% (*w*/*w*) PHB, dried BNC powder, 1% cinnamon leaf oil (*v*/*v*), solubilized in glacial acetic acid.	BPHB2
5% (*w*/*w*) PHB, dried BNC powder, 1% cinnamon leaf oil (*v*/*v*), solubilized in glacial acetic acid.	BPHB5
10% (*w*/*w*) PHB, dried BNC powder, 1% cinnamon leaf oil (*v*/*v*), solubilized in glacial acetic acid.	BPHB10
38.48% (*w*/*w*) BNC, 15% (*w*/*w*) polyethylene glycol (PEG), 15% (*w*/*w*) glycerol, 15% (*w*/*w*) Tween 80, 15% (*w*/*w*) PHA-rich biomass, 1% cinnamon leaf oil (*v*/*v*), solubilized in water	BPST
40% (*w*/*w*) BNC, 20% (*w*/*w*) glycerol, 20% (*w*/*w*) polyvinyl alcohol (PVA), 20% (*w*/*w*) PHA-rich biomass, 1% cinnamon leaf oil (*v*/*v*), solubilized in ethanol	BPSE
40% (*w*/*w*) BNC, 20% (*w*/*w*) glycerol, 20% (*w*/*w*) PVA, 20% (*w*/*w*) PHA-rich biomass, 1% cinnamon leaf oil (*v*/*v*), solubilized in water	BPSW

**Table 2 polymers-16-03490-t002:** Tensile properties of BNC and BNC/PHB blends.

Sample	Young’s Modulus at 0.2% (E) [MPa]	Ultimate Tensile Strength (UTS) [MPa]	Strain at UTS (ε_UTS_) [%]	Stress at Break (σ_B_) [MPa]	Strain at Break (ε_B_) [%]
BNC	1703.5 ± 221.5	27.9 ± 9.5	8.8 ± 3.8	27.8 ± 9.6	9.0 ± 3.7
BPHB2	912.3 ± 540.7	14.2 ± 7.1	4.8 ± 0.2	13.9 ± 7.5	5.0 ± 0.2
BPHB5	1173.7 ± 11.4	15.5 ± 3.6	3.6 ± 0.9	15.5 ± 3.6	3.7 ± 0.8
BPHB10	657.4 ± 126.5	6.9 ± 1.7	3.3 ± 0.6	6.7 ± 1.5	3.5 ± 0.5

## Data Availability

Data are contained within the article.

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
