# Peer review of "Development of Antimicrobial Blends of Bacteria Nanocellulose Derived from Plastic Waste and Polyhydroxybutyrate Enhanced with Essential Oils"

_polymers, 2024, doi:10.3390/polym16243490_

Round 1
Reviewer 1 Report
Comments and Suggestions for Authors Review report In this work, the authors aimed to prepare a two polymeric composite containing different oils as films and evaluate some physicochemical properties including morphology, structure and thermal properties, as well as their antibacterial activity to three microorganisms. Some issues need to be answered accurately: 1. The article is mostly written as a report of a laboratory work. It is suggested to refine the article similar to scientific articles. 2. It is suggested to specifically state the purpose of developing this system as well as its application area in the introduction. 3. In table 1, what are the compounds in the last three rows and where are they used in this work? 4. The items presented in 2.3.1 are based on what reference or references? Are there no other methods for reliable evaluation based on instrumental and not visual measurements to describe the morphology of films? 5. What was the purpose of the thermal test and what results did the authors seek from this test for the above work? 6. Why is FTIR test done in ATR mode? 7. Hasn't anything been done before in the field of evaluating the behavior of the oils used in this work against the mentioned microorganisms? If it has been done, why did the authors do it again? 8. What was the control sample in the inhibition zone test? 9. How is the name of the samples in Figure 4 done? 10. It is necessary to provide a typical curve of the examined samples in the tensile test and to compare the figures statistically, P-value should be used. 11. What is the meaning of lines 357-360? 12. It is necessary to discuss the mechanical properties of the samples: for example, why is the elastic modulus of BPH10 lower than that of BPH5? Or, in general, what range of mechanical properties can or should be suitable for this system for the applications proposed by the authors? 13. Based on what references, the intensity of FTIR peaks can be attributed to the concentration and integrity of the components? 14. In vertical axis FTIR curves, the word "arbitrary" should be replaced. 15. It is suggested that the results of the microbial growth of the samples be described and explained based on the structure of the microorganisms related to the physicochemical properties of the samples in order to determine the reason for the difference in the behavior of the samples, for example, against Gram-negative and Gram-positive bacteria. 16. It seems that the authors have implicitly mentioned the degradability of the samples for different applications, therefore it is suggested to evaluate their degradation behavior.
Author Response
Reviewer 1
- We appreciate the suggestion and have revised the manuscript.
- We have added a clear statement of the purpose of the research in the introduction, emphasizing the development of antimicrobial blends from recycled plastic waste for applications where hygiene and safety are crucial.
- We have removed the last three rows of Table 1, which represented BNC/PHA blends that were found to be brittle and unsuitable for further investigation.
- We have clarified the complimentary use of 2.5D morphology tool.
- We have clarified the purpose of the thermal test (TGA) and the results we sought, which was to evaluate the thermal stability and degradation behavior of the BNC blends.
- It’s employed due to its ease of use, minimal sample preparation, and suitability for solid and thin film samples.
- We have clarified the novelty of our research in the manuscript and added reference about the selected oil in the literature.
- We have clarified that the control in the manuscript.
- We have clarified the naming convention for the samples in Figure 4, based on their composition.
- We have included a typical stress-strain curve and performed statistical analysis using the p-value to compare tensile properties.
- We have clarified the rationale behind selecting 2%, 5%, and 10% PHB concentrations for the BNC/PHB blends.
- We have expanded the discussion on mechanical properties, including the trend in the elastic modulus and the suitable range for different applications.
- We have included references to support the interpretation of FTIR peak intensities concerning the concentration and integrity of the components.
- The equipment output related to the vertical axis label in the FTIR is "Transmittance %."
- We have expanded the discussion on antimicrobial activity, considering the structure of the microorganisms and the physicochemical properties of the blends.
- We have included a more explicit discussion on the biodegradability of the BNC/PHB blends and highlighted it as an area for future investigation.

Reviewer 2 Report
Comments and Suggestions for Authors
Authors reported nanocellulose from plastic wastes and PHB based materials reinforced with essential oils. The authors presented selected three compositions of biopolymer and tested the antimicrobial activity against three yeasts. The manuscript is well organized and plan to prove their proposed outcome. However, there are still unclear issues that need clarification so that the paper to be ready to be published. Please find below some comments/suggestions that may improve the quality of the manuscript.
1. The state of art of the topic was not convinsingly addressed. In the past 10 years at least there are numerous studies dealing with the antimicrobial films and coatings. The authors are requested to defend the novelty of their topic and to state how it is going behind the state of art.
2. The authors selected three compositions 2 ,5 and 10 % but no explanation behing their selection was included. The authors are requested to explain their selection.
3. The authors reported within in Figure 10 some negative values for antimicrobial activity which their statistical significance was not clearly understood. The authors are requested to provide details.
Author Response
- We have revised the introduction to more comprehensively address the state of the art of antimicrobial films and coatings and highlighted the novelty of our research.
- We have added clarification regarding the selection of 2%, 5%, and 10% PHB concentrations for the BNC/PHB blends.
- We have revised the discussion on the negative values for antimicrobial activity in Figure 10, explaining that they represent instances where the material increased biofilm formation compared to the control.

Round 2
Reviewer 2 Report
Comments and Suggestions for Authors
The authors answered to the addressed issues; however, the answer was simple without providing specific clarifications; more particularly on the negative values and their significance. The state of art is poorly highlighted. However, it is the responsibility of the authors to present valid results so that their work to be considered a reference for other studies.
Author Response
The authors answered to the addressed issues; however, the answer was simple without providing specific clarifications; more particularly on the negative values and their significance. The state of art is poorly highlighted. However, it is the responsibility of the authors to present valid results so that their work to be considered a reference for other studies.
Thank you for your feedback. We understand your concern about the negative values and the need for specific clarifications. We also acknowledge the need to further highlight the novelty of our approach. We have revised the manuscript to address these issues:
The section on "Determination of Anti-biofilm activity" has been significantly revised, both in the Methods and Results sections. The negative values encountered in the biofilm inhibition assay (Figure 11) were removed. These values indicated that the condition promoted microbial growth, which is not the main focus of the experiment. This ensures a clear understanding of the antimicrobial activity, which is the primary aim of this study.
We have further expanded the discussion on state-of-the-art to provide a more comprehensive overview of the existing research on incorporating essential oils into biopolymers for antimicrobial applications. Furthermore, we have highlighted how our approach uniquely contributes to this field by implementing antimicrobial properties to our plastic waste upcycling strategies, fitting the scope of this journal.
We believe that the revisions made to the manuscript have addressed the reviewer's concerns and provided the necessary clarifications.
